# Current Sensor Integration Issues with Wide-Bandgap Power Converters

**DOI:** 10.3390/s23146481

**Published:** 2023-07-18

**Authors:** Ali Parsa Sirat, Babak Parkhideh

**Affiliations:** Photovoltaic Integration Lab (PIL), Electrical and Computer Engineering (ECE) Department, Energy Production and Infrastructure Center (EPIC), University of North Carolina (UNC) at Charlotte, Charlotte, NC 28223, USA

**Keywords:** current sensing, WBG power converters, integration challenges, single-scheme sensors

## Abstract

Precise current sensing is essential for several power electronics’ protection, control, and reliability mechanisms. Even so, WBG power converters will likely struggle to develop a single current-sensing scheme to measure various types of currents due to the limited space and size of these devices, the required high sensing speed, and the high electromagnetic interference (EMI) emissions they cause. Analysis of existing current sensors was conducted in such terms with the objective of understanding the challenges associated with their integration into WBG power converters. Since each of these requirements has different design tradeoffs, it is challenging to consider one specific method of current sensing to be perfect for all situations; thus, the possibility of developing novel methods to improve the performance of these single-scheme current sensors is further explored.

## 1. Introduction

Power converters with outstanding power-density factors can be developed with the development of wide-bandgap (WBG) semiconductor switching devices, where size, weight, efficiency, and power density are optimized [1,2]. Due to their superior performance over those using Si MOSFETs or IGBTs, WBG converters are especially suitable for applications such as transportation, aerospace, and grid-connected technologies [3,4]. Specifically, these WBG semiconductor devices offer optimal performance in power-conversion applications due to their superior electrical properties. By increasing the switching frequency, the size of passive components (such as inductors, transformers, and capacitors) shrink subsequently, which results in more miniaturized and lighter converters (Figure 1a) [5]. Additionally, due to their fast switching speed and low switching losses, these devices have a higher power density, which makes them suitable for high-power applications. Moreover, their low leakage current and temperature-independent switching characteristics make them suitable for a wide variety of applications. By reducing losses, the necessity for heat dissipation decreases. This indicates that a lighter cooling system can be applied, and accordingly, the overall size, weight, and efficiency will be improved. Generally, a WBG converter has a better size, weight, efficiency, and power density than an IGBT or Si Mosfet converter [1,2,3,4,5].

As opposed to old-fashioned converters, WBG converters require a greater degree of complexity for design and manufacturing [6,7]. Due to tight requirements, which are primarily concerned with size and bandwidth (BW), it is more difficult to integrate components, such as sensors. Generally, the ability to precisely measure electrical currents is essential for several protection, control, and reliability mechanisms within power converters. Even so, it is likely to be challenging for WBG power electronics converters to develop a single current-sensing scheme to measure various types of currents incorporated into WBG devices due to the limited geometry and size of these devices, the required sensing speed, fluctuating ambient temperature, and the high electromagnetic interference (EMI) emissions they cause [8].

This review further explains that there is almost no specific current-sensing method that provide all the necessary size, BW, linearity, isolation, accuracy, and cost to be considered complete, in one place. This is because each of these requirements has different tradeoffs for design considerations. As an example of characteristics, accuracy may come at the cost of size, while noise immunity may require increased complexity. Hall-effect (HE) sensors are an example of different existing schemes that do not have adequate accuracy or suitable BW [9] and, along with magnetoresistive (MR) sensors, they need to be compensated for temperature drift. For instance, an HE device must be compensated for temperature through closed-loop sensing and complex signal-conditioning circuitry [9,10]. Current transformers (CTs) are commonly known for being bulky and becoming magnetically saturated by observing amplitude-significant currents or magnetic fields [11]. Consequently, this saturation can adversely affect the CT’s performance and accuracy, making it difficult to rely on the readings from these devices. Even though Rogowski coils have proven to be a very good option regarding effective volume, cost, and high-frequency sensing, they are not able to detect DCs (as is the case with CTs since DCs have zero *di*/*dt*) [11,12]. Therefore, a second sensor capable of detecting DCs, such as an HE or MR sensor (or any other applicable magnetometer), is necessary. Shunt resistors can be used to measure the current of a circuit from DC up to MHz (or GHz) frequencies, but they may cause problems regarding their isolation, invasiveness, and temperature-dependent sensitivity, among others. For example, a shunt resistor’s measurement accuracy can be affected by the temperature of the resistor as the resistance of the resistor increases with the temperature [11]. As well as providing electrical isolation, fluxgate (FG) and magneto-optical (MO) schemes offer minimal temperature drift, high accuracy, and immunity to EMI noises, and they are particularly advantageous in low-frequency applications, such as measuring DC or low-frequency AC signals [9]. Despite this, implementing these systems is like building a very large, intricate, and expensive circuit by joining together fragile, interlocking components. It is true that they can offer excellent performance, but the cost of implementing them is often too high to make them economically viable [9]. Nevertheless, recent technological advances have made it possible to fabricate on-chip systems that are more efficient and less costly [13]. 

A study of existing single-scheme (SS) current sensors’ performance is presented in this paper to understand the challenges associated with incorporating them into the architecture of emerging WBG power converters, along with their specific applications in protection, control, reliability, and characterization as shown in Figure 1b, where the thicker arrow shows the greater impact of each factor based on the application. Additionally, the potential to develop novel methods to improve the performance of these SS current sensors is also explored. For instance, the survey evaluated the performance of single-scheme current sensors in terms of their accuracy, linearity, and dynamic range, as well as the influence of high EMI noise on their performance.

## 2. Current-Sensing Methods in Power Applications

A current sensor is a device exploited to measure electric currents, either alternating current (AC) or direct current (DC). These devices are used in a variety of applications, including detecting faults, monitoring performance, and detecting changes in system behavior. Basically, power applications employ a variety of current-monitoring products to control, characterize, and monitor the current flow [14]. Current sensors, for example, are used in power lines to measure the flow of the current, which can detect overcurrent or excess currents [15]. Furthermore, current monitoring can be utilized for several purposes, such as fault detection, monitoring performance, and observing changes in system behavior [16,17]. In this section, single schemes of current-sensing methods are categorized into four branches [9] and their working principles are briefly discussed. Afterwards, their integration issues within power electronics are analyzed. The four categories of sensing technologies are as follows: (1) ohmic current sensors; (2) DC magnetic-field detector; (3) inductive-based sensing schemes; and (4) magneto-optic sensing technology. 

### 2.1. Ohmic Current Sensors

By passing the current through a conductor, the voltage drop across the conductor can be measured for current-sensing purposes [18]. As shown in Figure 2, knowing the impedance of the current-sensing element can yield the current vector via dividing the measured voltage by the impedance value. This type of current-sensing mechanism, which is based on Ohm’s law, can be considered as a resistive or ohmic current-sensing principle. All shunt resistors (SMD or coaxial), copper trace method, MOSFET sensing (using either sense FETs or on-resistance of the MOSFET), and inductor voltage sensing (in which an integrator is needed to convert the inductor voltage to its current by assuming a constant inductance value) schemes are classified in this category [9,18]. 

In resistive methods (such as coaxial shunt), an ultra-high BW can be achieved (even up to a GHz [19,20]) by lowering parasitic inductance techniques. Their other advantages include simple passive circuitry (except the signal-conditioning part), small size, and cheap cost. Still, a significant problem of these schemes is connection (or placing the sensing element in series with the current trace), which also lacks electrical isolation, introducing parasitic elements (such as series inductance) to the power traces, and creates power loss and heat (that can also limit the current-sensing capacity of the element) [19,20,21]. One another big issue of ohmic current sensors is sensitivity change due to temperature variations [9], which is related to the temperature dependency of impedance values. Particularly, metal resistance can vary with any change in its temperature. In addition, as an ohmic current-sensing method, one technique of reading the inductor current is to read the inductor voltage and integrate it. This type of current sensing can also be affected by temperature variations in the inductor core.

### 2.2. DC Magnetometers

Magnetic sensors have evolved from ancient navigation devices to meet requirements for increased sensitivity, smaller sizes, and compatibility with electronics, for a wide range of applications, including current measurement [22]. The intensity of the magnetic field or flux density (*B*) (as the target of these sensors to detect) can be directly converted to the intensity of the electric current (*I*), based on Ampere’s law (1), in the context of current sensing:*I* = 2*πrB*/*μ*,(1)

In which *r* is the distance from the current-carrying conductor, and *μ* is the magnetic permeability. Furthermore, these sensors can now be easily integrated with other semiconductors, which is possible due to advances in miniaturization and manufacturing processes, where they have allowed for smaller and more efficient components that enable the sensors to be more accurate, reliable, and responsive [23]. However, there are some drawbacks to magnetic sensors, for example, they can be affected by outside magnetic fields, and they also require a power source [24]. Three widely used magnetic sensors for current sensing in power applications are as follows: (1) Hall-effect (HE) sensors; (2) magnetoresistive (MR) sensors; and (3) fluxgate sensors (FG). 

#### 2.2.1. Hall-Effect Sensors

HE sensors are based on the Hall-effect phenomenon that occurs across holes or voids in semiconductors or metals when a current is injected through contact near the edge or boundary of the gap [25]. Consequently, a voltage is produced on either side of a line connecting the current contacts when a perpendicular magnetic field is applied outside the gap within the metal or semiconductor material. Hence, HE detectors determine magnetic-field intensity according to the current sensed through the input interface, which must be subjected to signal conditioning to produce a significant output. Hall voltages are low-level signals (that can reach a maximum of a few dozen microvolts when surrounded by one gauss magnetic field), so an amplifier with low noise, high impedance, and moderate gain is required [26]. If the signal conditioner is not optimized for the application, it may not provide the highest level of accuracy. Due to interference from external fields, magnetic cores with high permeability can reduce interference effects by maximizing and concentrating magnetic fluxes from the measured current. As shown in Figure 3, the HE schemes can be implemented in either an open-loop or a closed-loop system, and depending on the needs of the application, they can be either with a magnetic core or they can be coreless. Closed-loop systems can help reduce nonlinearities by providing feedback to the system and adjusting the parameters accordingly, which allows the system to compensate for thermal drift. Furthermore, it is important to note that HE schemes typically have response speeds way lower than 100 kHz, and adding a CT configuration to the system can improve the sensing BW in the closed-loop system [9]. 

#### 2.2.2. Magnetoresistive Sensors

The concept for MR sensors was first reported by Hunt in 1971, though magnetoresistance was discovered by Lord Kelvin in 1857 [27]. MR sensors can detect changes in both the strength and direction of the magnetic field by adjusting their resistance, which makes them suitable for a variety of applications, such as current sensing. Because of the special material used in making these sensors (usually a mixture of iron and nickel alloys), in addition to being exceptionally small and robust, they consume very little energy [28]. The anisotropic magnetoresistive (AMR) and giant magnetoresistive (GMR) sensors have attracted more attention among the various types of MR, owing to their ease of use and physical characteristics [22]. Other types of MR sensors, such as tunneling magnetoresistance (TMR) sensors, are usually more accurate and precise than AMR and GMR sensors [28], and due to their small size and outstanding performance, TMR sensors are now becoming more popular for power electronics applications. To enhance the linearity of the MR sensor and compensate for any thermal differences, MR sensors are usually installed within a Wheatstone bridge, which enhances the linearity of the sensor [29]. Figure 4 depicts the physical structure of three different MR sensors.

#### 2.2.3. Fluxgate Sensors

FG technology sensors take advantage of magnetic core saturation, whereby passing a current close to the sensing core and measuring the flux, the core can be saturated and then unsaturated to measure the relative current that causes the magnetic field [9,28]. Parallel FG sensors are characterized by parallel excitation fields, whereas orthogonal FG sensors have excitation fields perpendicular to the sensitivity axis [28]. In Figure 5, an example of a parallel FG sensor can be seen. A sense coil is coupled with a signal-conditioning circuit, as well as a compensation coil, so that closed-loop control can be implemented within the circuit, in which the sensor output is integrated to achieve high loop gain. From a technical perspective, the integrator is connected to the differential driver that runs an opposing compensation current out of the inner compensation coil. By generating an inverse magnetic field in the compensation coil, the sense coil’s initial field returns to zero, where the compensation coil constantly generating a magnetic field and nonlinearities in the field caused by the input signals are compensated for. With outstanding accuracy, FG elements are continuously propelled in and out of saturation, maintaining zero-hysteresis control, and the shunt resistor senses the current produced by the external current in proportion to that produced by the external field, ensuring steady gain and high linearity. An FG sensor is typically considered to be the most accurate magnetometer among all the different magnetic-field detectors available today [9,28]. However, FG sensors have certain limitations, such as complex control circuitry, which was traditionally a major problem when considering their size and makes the scheme relatively expensive in comparison with other types [9], but as a result of the coming-of-age of solid-state technology, IC-level FG sensors can solve the problem of size and cost [13].

Several advantages are associated with magnetic-field detectors being utilized as current sensors, including DC measurement capability, ease of application, low cost, and good isolation, while the disadvantages of these sensors typically include their low BW, interference with external magnetic fields, their susceptibility to EMI noises (especially when used in conjunction with metal plates that sense), and their limited sensing range, which must always be compromised when integrated into power electronics. 

### 2.3. Inductive Current Sensors

Coils, whether they have a magnetic core, can always be subjected to a passive or electronic conditioning circuit to measure the alternative current (AC) using Faraday’s law of induction (2) [11,12]:*Induced Voltage* = −*M*
*di*/*dt*,(2)
in which *M* is the mutual inductance between the coil and the conductor. From (2), induced voltages in higher-frequency ranges are larger than those in lower-frequency ranges, which means that this type of scheme is more accurate at higher frequencies (although only up to the coil’s resonant frequency, parasitic values will dampen the induced voltages at frequencies higher than the resonant frequency [30]), and there is no induced voltage when the coil is in a DC or a static magnetic field (*di*/*dt* = 0). By increasing the number of turns in the coil, the induction voltages are also increased, which will again affect the final BW to be decreased [12]. This is because the greater number of turns in the coil will also boost the overall inductance of the coil. In power applications, current transformers (CTs) (with a magnetic core and all passive circuits) and Rogowski coils (RCs) (employing electronics to compensate for the low gain of a core-less coil) are two common sensing schemes of this type [9,11,12,30].

#### 2.3.1. Current Transformer

The CT is a type of coil-sensing realization that consists of three main parts, namely a coil or winding, a core, and a termination. The number of turns of the coil and the magnetic properties of the core can provide information regarding coil physics and some other electrical properties. CT sensing is versatile, as the number of turns and core properties can be adjusted to better suit various sensing requirements. The equivalent *RLC* of the coil can be calculated using coil and core physics, as shown in Figure 6, allowing for the determination of the upper and lower bands of working frequencies [11]. A resistor can be deployed at the termination of each CT for (*RL*) self-integration purposes, allowing it to be modified separately for a specific frequency range. For example, the resistor at the termination of the CT can be adjusted to shift the upper-frequency limit from 10 kHz to 100 kHz [31]. While CTs are ideal for many power applications due to their noise immunity, low cost, isolation, and passive circuitry, its integration has several drawbacks, including core saturation (limiting the sensing capacity and accuracy), fluctuations in core permeability at variable temperatures, low permeability at high frequencies, and bulkiness associated with having a magnetic core. However, for instance, to reduce the size of CTs while increasing their sensing ability and accuracy, nanocrystalline cores [32] are used instead of traditional magnetic cores.

#### 2.3.2. Rogowski Coil

Although the RC working principle is very similar to CT, they usually use air-core coils instead of magnetic cores, so they generally require electronic integrator circuitry to compensate for the lower gain provided by an air core (especially at low frequencies) [30]. Since air-core coils do not experience magnetic saturation, they are often preferred in RC circuits because of their sensing capability, which is not as limited as it is with magnetic core coils, and they can also work at higher frequencies [31]. Air-core coils’ smaller dimensions and volume make them even more appealing for use in RC circuits because of their superior performance. In Figure 7, there are three major components of an RC current sensor: 1, coil; 2, termination; and 3, integrator [12,30,31]. Based on the RC’s frequency response, it can sense ultra-high frequencies, but like CT, it cannot detect DC/very low frequencies. 

Technically, inductive-based current sensors are good AC transducers and are the only options to non-invasively measure high-frequency currents. In addition, they provide isolation due to their physical characteristics, and their greatest disadvantage may be their incapacity to detect DC. In Figure 7b, a typical value for *F_a_* is usually Hz~kHz, while for *F_b_* it is usually 100 kHz~MHz, and for *F_c_* it is usually MHz~GHz.

### 2.4. Magneto-Optical Current Sensing

Magneto-optical (MO) current clamps, which utilize Faraday rotation when polarized light crosses an MO material in a magnetic field, have potential applications in power converters, where fiber-optic-packaging techniques can make MO current sensors relatively small, and high sensitivity can be achieved using rare-earth ion garnet materials [33]. In addition, MO current sensors are intrinsically immune to EMI noise and temperature variations. However, there are some potential challenges to implementing MO current sensors. One challenge is that MO materials currently available have limited bandwidth [9], which means that they may not be able to accurately measure HF currents. The cost effectiveness of MO current sensors may be limited for some applications [9], which means it may not be feasible to employ them to measure very low currents. An MO current sensor application within a power electronics circuit is shown in Figure 8.

## 3. Current Sensor Integration Concerns in WBG Power Electronics

All requirements of the power converter’s current measurements obtained by the sensors should be of sufficient quality to ensure successful integration into these systems, where accuracy of the protection, control, and monitoring methods in power converters directly depend on the quality of the in-situ current measurement [31]. The development of a single current-detection scheme suitable for power electronics (especially WBG devices) that can accurately detect a wide range of current types will be extremely challenging due to the restricted geometry/size, the demand for fast sensing speeds, variations in ambient temperatures, and the excessive EMI radiation levels [14]. For example, the high frequencies of modern semiconductor devices cause immense levels of EMI [34], which can lead to false detection and inaccurate measurements from the current-sensing scheme. There have been many current-detection schemes developed, but in most cases, they are not of the highest quality, and their use in power converters need to be compromised. Several key concerns related to the integration of current sensing into power converters are introduced, and the methods described in the previous section are critically analyzed based on those concerns in this section. This assessment considers the following factors: (1) isolation; (2) invasiveness; (3) size and bulkiness; (4) BW and DC measurement capability; (5) switching EMI noise immunity; (6) accuracy; (7) sensing range capacity; (8) thermal drift; (9) power consumption and losses; and (10) cost.

### 3.1. Electrical Isolation

Apart from ohmic (resistive)-based current-sensing schemes, most current-sensing schemes are intrinsically isolated, which does not require a direct electrical connection [18]. A direct electrical connection is not required for inductive, optical, or magnetic-field current-sensing schemes [9], which permit isolation and safety. Using resistive methods (such as coaxial shunt current sensor), the primary circuits are directly connected to the sensing circuits, which may not be a major concern in low-voltage electronics; however, it is of greater concern in power electronics, where the common mode voltage difference may be significant between the various circuit components, highlighting the absence of electrical isolation. The use of shunt sensors or any similar sensor technology, such as sense FETs, is therefore restricted to low-voltage applications or low-side switch-current monitoring in power electronics. Sense FETS can, for example, be used in low-voltage converters as a shunt resistor to measure current by connecting in parallel with the low-side switch, while high-side modified sense FETS can be used in low-voltage converters as well, but [35] illustrates that isolation requires more complex circuitry. To monitor switch current in double-pulse testers (DPTs), a (coaxial) shunt resistor could be placed in series with the low-side switch [19], allowing for accurate current monitoring without the need for isolation, such as in Figure 9.

### 3.2. Circuit Invasion

Being invasive can be described in terms of the need to pass current electrically through a current sensor to measure it, thus invading the electrical circuit to measure the current. A current sensor of this type is considered an invasive current sensor, and for instance, a shunt resistor is an invasive current sensor, as the resistor must be placed in series with the circuit. It is also possible to measure current using non-invasive or contactless sensors instead of inserting a probe into the circuit as an alternative, which provide a means to measure current without interfering with the electrical circuit. A contactless current sensor may, for example, employ MR technologies, where the idea is that a current flowing through a conductor will create a magnetic field that can be detected with a magnetometer. For current-sensing (especially at a high voltage) applications requiring safety and accuracy, current clamp (or non-invasive) detectors are the ideal choice, which do not need to alter the optimal power layouts of power electronics (especially those based on WBG). As an example, a high-voltage three-phase inverter utilizing a PCB-embedded RC current sensor can maintain the ideal power layout of WBG semiconductor devices while still providing accurate current-measurement capabilities [31]. 

Resistive current-sensing methods present significant challenges due to their electrical contact. Besides introducing parasitic elements to the power traces (such as series inductance), these methods can also result in heat loss and power loss (which can limit the element’s capacity to sense current) [9,18]. While these sensing schemes have been used in a number of specific applications, including DPT for characterization (because of their excellent sensing bandwidth, as shown in Figure 9), medium-frequency (MF) converters, or small power/voltage applications [36], they are unreliable/integrable for medium/high-power converters, due to the absence of isolation. Additionally, parasitic elements are introduced and nonlinearity is present (of temperature variation). In Figure 9, an example of the coaxial shunt from [19] is shown to illustrate the issues relating to the integrability of ohmic current-sensing approaches within WBG (GaN-based) power electronics converters. This type of shunt is composed of two concentric conductors with a dielectric material in between. This optimized coaxial shunt has a bandwidth of approximately 1 GHz, but it is implemented in series low-side switches to reduce isolation problems. The current waveform illustrates that too much ringing was generated during the switching transient because of the invasive shunt resistor, implying the introduced parasitic values’ impact.

Inductive, magnetic field, and magneto-optical current-sensing methods are intrinsically capable of detecting currents using electromagnetic fields without an electrical connection [9]. Despite this, some integrated circuits (ICs), such as HE IC sensors, require a current to be passed through them [37] to concentrate the magnetic fields, which is mainly because HE elements are not as sensitive to small magnetic fields as MR or FG [28]. The issues related to invasion, such as altering the power trace path and introducing parasitics, may limit their use within optimal power-converter layouts [38], even though they can provide isolation. MR or FG technologies, which are largely non-invasive, may prove more appropriate for use within an optimal power-converter layout if their design does not incorporate significant parasitic values. Moreover, CTs (or other methods) with a large magnetic core may add too much inductance (usually bigger than one micro-Henry) to the primary circuit, even when they are non-invasive electrically, which is known as inductance insertion [39]. It could be argued that PCB-embedded or RC probes would make better alternatives since they are completely non-invasive and their inserted inductance is very small (usually not more than a few nano-Henry). 

### 3.3. Size and Bulkiness

The size and volume of a current sensor are also critical for integration into optimal power-converter layouts, in addition to being isolated and non-invasive, where keeping the size and volume of a sensor to a minimum allows for greater flexibility in the design and additions to the layout, which reduces the cost and complexity of the overall system [39]. Figure 10 displays a visualization of existing current sensors or probes. Having smaller dimensions offers greater integration flexibility and allows for a greater number of sensors to be integrated since a smaller volume makes it easier to install and maintain sensors. CT, for example, offers excellent isolation in a non-invasive manner; however, its bulky magnetic core (number 3 in Figure 10) may not be adequate for monitoring switch currents [40,41], where the provided illustration of this issue in Figure 11 displays the height of the electric power trace, as well as the length of the overall power trace in the application, cause too much ringing during switching, which implies excessive inductance insertion. Also, the non-invasive closed-loop HE sensor [42] shown as number 2 in Figure 10 illustrates why incorporating a magnetic core into the sensing structure might not be feasible for small-size converters. Furthermore, it is also important to point out that WBG converters cannot be equipped with current probes (number 1 in Figure 10 [43]) for control or protection because of their immense size and cost. There are several different magnetic-field-based current sensors, but MR sensors (specifically AMR and TMR) are contactless magnetometers that are small-size ICs [44] (such as number 7 in Figure 10), however, coreless HE ICs are usually invasive [45]. Due to its PCB/air-core technology (numbers 4 and 8 in Figure 10), RCs offer an easier alternative to bulky inductive-based circuitry when integrating into a WBG converter, which are very compact and can also be embedded within power board or gate driver layouts. It has also been reported that magneto-optical (MO) sensors and probes contain sophisticated circuitry (Figure 8) that could make them too large, and this could be the same for FG probes in general [9]. Despite this, the on-chip FG solved the size issue (number 6 in Figure 10) [13,46]. 

### 3.4. Bandwidth and DC Measurement Capability

The current-sensing bandwidth refers to the range of frequencies over which a sensor can accurately measure current, and its requirements can vary depending on the application and the specific type of current or equipment being measured. Typically, higher BW current sensors are required in WBG power electronics due to the faster switching frequency and sharper turn-on/off characteristics [14,30,47,48,49,50,51,52], where the BW should be set wide enough to capture the HF components of current waveforms [11,12]. As well, the switch-current waveform can be square/trapezoidal with frequency components ranging from DC to extremely high frequencies [12], and the only single-scheme current sensor capable of measuring DC up to tens of MHz (or GHz) is the coaxial shunt [20], which has integration problems relating to isolation and invasiveness. A diagram in Figure 12 illustrates how potential measurement BWs of SS current sensors may be explained, in which, for example, a CT may be modified to suit the low-frequency or HF applications [11]. Although both inductive methods (RC and CT) can reach high frequencies in the upper-frequency band, they cannot detect DCs [9,11,12,18,30]. On the other hand, magnetometers (HE, MR, and FG) and MO, which perform well at DC and low frequencies (LFs), are unable to construct a very high BW due to their physical structures [9,28]. On-chip or micro-FG devices typically limit the excitation signal to 10 to 100 kHz because of the parasitics of the coil and the high magnetization of the core [13,28,53,54]. Despite the limitations of each sensing method, magnetometers, inductive methods, and a combination of the two offer a variety of viable options for measuring current in a wide range of bandwidths. A combination of multiple sensing methods can be used to detect a wide range of frequencies [14,55], a topic that is addressed later in the discussion.

### 3.5. Switching EMI Immunity

One of the most significant obstacles to sensing and signal conditioning in WBG power electronics is the requirement for integrated current sensors capable of handling significant *di*/*dt* and *dv*/*dt* (Figure 13). Due to their HF switching and ultrafast turn-on/off, they can be subject to high *di*/*dt* in switching commutation loops that can cause ringing and overvoltage, resulting in RF emissions, EMI radiation, and additional losses, as well as spurious gate voltages, resulting in high coupling due to large *dv*/*dt* [56,57,58]. These issues can interfere with the operation of other components in the system and reduce overall system performance and reliability. For instance, improper turn-off snubbing can lead to a resonance between the bridge leg inductance and the stray capacitance, resulting in destructive transients. These transients can cause circuit malfunctions and harm the system’s performance. To mitigate these issues, proper design of sensing circuitry and optimal layout of the converter can reduce the problems associated with high *di*/*dt*, which is also emphasized in Figure 14 [59]. 

Except for the ways to conduct EMI that can be easily eliminated by isolated sensing circuits, it can be coupled near the field or radiated, in which EMI noise can be coupled or induced by parasitic inductances and capacitances within the sensing circuit [59]. The amount of coupled noise currents can be reduced by minimizing stray capacitance (by shrinking the conductor connections and increasing their distance from each conductor), and to minimize the induced EMI noise voltage, the parasitic inductance of these sensing circuits must be reduced [60,61,62,63,64]. 

The high radiation potential of EMI is also a reason why non-isolated or invasive current sensors are better not utilized as parasitics introduced by them into converter layouts can contribute to increased EMI issues, and their considerable amount of conducted and coupled noise can interfere with the system’s protection and control systems. The use of inductive current sensors from an EMI standpoint requires avoiding high inductance insertion into converter layouts, as well as decreasing their stray capacitance to avoid the requirement for shielding (for BW), so either very small CTs or RCs embedded on PCBs with very excellent EMI immunity can be implemented [11,12,18,30]. Since HE sensors are either invasive or large/bulky, they do not provide an appropriate representation of magnetometers in terms of EMI immunity. Furthermore, low-pass-filtered outputs are also necessary when non-invasive AMR or TMR ICs are placed near switching nodes (because of coupled and radiated EMI), which further restricts their BW. Figure 15 illustrates that, while TMR has superior sensitivity and BW [28], it is prone to HF switching EMI, and its output must be cut off with a low-pass filter (LPF) at a very LF (usually less than 50 kHz) in order for the output to remain functional. 

### 3.6. Accuracy and Linearity

The accuracy of a current sensor is determined by how closely it can measure the precise value of the target current in the time or frequency domain. There are numerous factors that can affect sensor accuracy, such as temperature, magnetic hysteresis, and electrical noise. Several of the concerns associated with current sensor integration could also be associated with reading accuracy, such as BW, thermal drift, and immunity to EMI noise. For instance, CTs or RCs may not accurately sense if the current has DC or very low-frequency components [30]. Conversely, FG or HE may not be capable of detecting very fast-rising currents accurately due to their low BW. In addition, fluctuations in ambient temperature may cause nonlinear readings (in resistive current sensors [9,11,12]) or thermal drift (in magnetic sensors [28]), which affects the accuracy of the measurement. Temperature changes alter the resistance value of shunt resistors, which leads to inaccurate and nonlinear measurements [9,18] where they are prone to be warmed up by passing large currents through them, also changing their sensitivity. In CTs, due to the saturation of their magnetic cores, as well as hysteresis in the case of very large currents or magnetic-field changes, there are nonlinear readings and a limitation on their capability to provide accurate sensing [11]. However, RCs do not contain a magnetic core, so, they provide excellent linearity compared to CTs. The PCB-embedded RCs are therefore an excellent choice for measuring HF currents since their differential versions are not affected by external magnetic fields, making them an inexpensive, accurate (if the DC lack is compensated), and reliable solution [12,18,30,32].

Except for FG (which can be considered as the most linear and accurate magnetic-based current sensor [9,13,28,54]), most magnetic-field detectors have linearity limitations due to their physics (especially HE and AMR [28]). Moreover, the output of a current sensor should be free of noise when it comes to accuracy, as noisy readings tend to indicate inaccurate measurements. Other than open-loop HE or MR sensors that are prone to thermal drift (which affects their sensing accuracy), small contactless MR or FG ICs can be low-pass-filtered to provide reasonably noise-free outputs (Figure 15 and Figure 16). As shown in Figure 16, an LPF can be employed to decrease HF noise and increase the accuracy of the output signal of the micro-FG (DRV425 [61]), which can be implemented with a simple RC (resistor–capacitor) network; however, this will reduce the output signal BW. Finally, offset drift (that is usually caused by magnetic hysteresis) is another phenomenon that could affect current-sensing accuracy, especially in magnetic-field detectors. For instance, the offset drift of an HE sensor (known as one of the poorest magnetometers in this term [28]) can result in false readings that cannot be related to the actual current, resulting in serious accuracy and reliability problems. Not only does it have the best noise immunity and accuracy, FG has the lowest offset drift among other magnetometers, and TMR can be considered as an alternative (with a higher offset drift) [28].

### 3.7. Thermal Drift

As a result of temperature fluctuations, thermal drift occurs when the current sensor’s output is different, almost under the same electrical conditions, when the variations are not adequately compensated. The output voltage of ohmic current sensors (shunts), for example, increases with the temperature due to increased resistance, resulting in inaccurate measurements in temperature-variable systems [9]. For example, a closed-loop current-measurement system, such as FG, MO, or RC (with an analog integrator) can compensate for any thermal variation, due to their intrinsic feedback system [9,30,32,66]. In contrast, current-sensing schemes, such as open-loop MR or HE, can be highly susceptible to such fluctuations. To avoid this, temperature compensation can be implemented in the sensor to ensure accuracy [67,68]. The unit to quantify the thermal drift of a current sensor is ppm/K, and any value below 100 ppm/K is considered acceptable [9]. FG is considered by many to be the best non-invasive current sensor when it comes to thermal drift [9,28], although MO, RC, and CT can also achieve acceptable thermal drift values under the right conditions. FG sensors offer superior thermal drift characteristics because they are not affected by temperature as much as other non-invasive sensor technologies, such as MO, RC, and CT. Furthermore, they are very accurate, which is why they are the preferred choice for many applications. Due to this consideration, closed-loop AMR and TMR [28] can have acceptable levels of thermal drift, as well as closed-loop HE [69], within a reasonable range. 

### 3.8. Power Consumption

The power consumption of a current sensor may be caused by conduction losses (such as resistive power loss in invasive sensors) or by powering up its processing circuits (such as power supplies in contactless magnetic-field sensors). It is important to consider the power consumption of sensors for specific applications (Figure 17), especially when they are susceptible to excessive heat or are located inside high-efficiency low-power converters [69]. In such terms, low-power sensors are more convenient to integrate since they require minimal power supply and do not require a cooling system, thereby making installation and setup easier. While FG ICs have the highest sensitivity, accuracy, and temperature insusceptibility of all the magnetometers, it also has the highest power consumption among them [28,69]. The reason for this is that micro-FG sensors require compensation current to equilibrium magnetic fields within its magnetic core to attain a high degree of linearity. Since on-chip FG may need high powers, they are not suitable for loss-limited conditions, particularly in applications related to the Internet of Things (IoT), but according to [54], they designed and implemented an FG IC with only 13 mW power consumption for those applications.

### 3.9. Sensing Range

This term refers to the maximum and minimum currents that can be accurately sensed by the sensor. The accuracy of the measurement is affected if the current sensor measures currents outside the linear operating range (also known as the linear operating range). For instance, a current sensor with a linear operating range of 0–100 A would not be suitable for a bidirectional application with a current reading requirement of −50 to +50 A. Resistive current sensors have linearity limitations because of heat dissipation, which increases resistance as the current passes through the sense element [9,69]. Core saturation is another linearity limiter in magnetic-based current sensors. Core saturation occurs when the magnetic field generated by the current passing through the core reaches a certain level, and the core becomes saturated with magnetic flux, which limits the current that can be accurately measured, resulting in a nonlinear sensing condition [11,69]. In other sensing mechanisms such as RC or FG, voltage supply is the sensing range limiter. RC sensors are almost always linear unless they reach their opamp supply voltage. With an RC that has a sensitivity of 10 mV/A and an opamp supply voltage of −5 V to +5 V, the sensing range is limited to −500 A to +500 A [30], which is still significantly greater than the sensing range of a CT of the same size. Regarding the minimum current-sensing range, it is highly dependent on the sensitivity and noise immunity of the current sensor to detect very small currents/magnetic fields, which can once again be a matter of accuracy. It is also dependent upon the ambient noise levels in the environment, as well as the type of current sensor used to be immune to that level of noise. Even very small currents can be detected by high-resolution current sensors due to their high resolution, such as FG detectors [28]. In a nutshell, RC, MO, and FG, along with some coreless MRs or HEs, are viable candidates for very high maximum-current-sensing capacities. With the right current sensor, even the smallest of currents can be detected, allowing for more accurate readings. Shunts, FG, TMR, and AMR can be considered in the case of very small currents [28,69]. There are several microelectronics current-sensing level sensors presented in [70], but most of them have no practical application in power electronics.

### 3.10. Cost

One major concern with current-sensor selection with power converters is their cost. Sensor technology is rapidly advancing, allowing higher accuracy and lower costs. For example, MEMS sensors have become increasingly popular due to their small size, low cost, and high precision. As shown in Figure 18, developing FG sensors on the chip results in significant cost savings compared to FG clamps, and this is also true for the development of HE technology, which also results in substantial cost savings along the way. However, there is no commercial counterpart for RC probes on the market to replace them as smaller RC sensors. However, advancements in integrating RC as a PCB-embedded coil show the implementation feasibility of their implementation with simple circuitry [12,30,32,71,72,73,74]. Figure 18 illustrates the cost diagram based on up-to-date market data (accessed on 10 June 2023 from Digikey.com [71]), which can also be compared with the information found in [9,28,69]. According to the information provided, the most cost-effective contactless current-sensing schemes are the CT, embedded RC, micro-FG, and MR technologies.

## 4. Current Sensor Selection Based on Application

A thorough understanding of power electronics, current sensors, and utilization algorithms is required to integrate current sensors into power converters, where engineers should contemplate the complex relationships among these components to ensure they meet specific requirements and function properly [14], which will require detailed knowledge of the application and its necessities. The purpose of this section is to introduce a set of applications that employ current sensors in power electronics, along with sufficient details regarding suitable candidates for each application. There are four categories of applications that are examined in this research, namely protection, control, prognostics, and characterization [23,30,38]. Different types of sensors are recommended for each application based on their characteristics and features. For example, current sensors designed for protection should have a super-fast response time, while current sensors intended for control must be super accurate [32,38].

### 4.1. Protection

For effective protection against destructive currents, a suitable current sensor is usually required to detect faults when they occur. A current sensor must be able to detect fault currents accurately at high speeds, because the faster the current sensor can detect a fault, the quicker the protection system can act to trip the circuit and prevent further damage from happening. Furthermore, the device must also be capable of handling very high levels of current since fault currents can be very high at times. As well as this, if the current sensors are to be utilized within power electronics protection mechanisms, all other integration concerns that have been discussed previously must be addressed. Despite this, current sensors could have some drawbacks. For example, if the current sensors are not properly calibrated, they can cause false readings that could trip circuit breakers unnecessarily or cause other problems. Current sensors, however, can provide the best interpretation of faults associated with overcurrents and short circuits, which illustrates the importance of calibrating them correctly [32,38]. 

There is an increasing use of solid-state circuit breakers (SSCB) in a variety of emerging electrical systems, where several power electronics devices perform at higher speeds than electromechanical mechanisms typically employed by conventional circuit breakers and interrupt fault currents without moving parts [75]. Usually, these devices detect fault currents by sensing them directly or indirectly. The fault current sensor must have a high BW (which also means a fast response time), low loss (or no loss), and the ability to be integrated into the tripping circuit. Based on the circuit inductance, short-circuit faults can also result in wildly fluctuating current rates [30]. To maintain the highest level of breaker efficiency, the sensor should have a fast response time and lower losses than the main power semiconductor [75]. As shown in Figure 19, the literature describes the SSCB key components, as well as the mechanism for sensing and tripping. Since a wide range of voltage and temperature changes can cause a false fault to be inferred from indirect sensing methods, they may not be as reliable as direct current sensing. In [76], desat is compared to embedded RC short-circuit detection to illustrate the differences in accuracy and speed between direct and indirect fault-detection methods.

Several upcoming direct-current-sensing methods, including TMR and micro-FG, can potentially replace HE and GMR soon, as described in the previous section. A CT is generally not suitable for this purpose, mainly because it suffers from core saturation, which makes it difficult to sustain high currents for an extended period. However, embedded RC appears to be a good candidate, particularly if its SSCB fault-detection mechanisms do not require DC sensing. MR and RC (along with their hybrid combinations) have been underutilized extensively when it comes to short-circuit and overcurrent protection for WBG power converters [15,30,32,74,76,77,78] due to their high BW, small size, low cost, and non-invasive nature. 

### 4.2. Control

Many of the control schemes that are used in the field of power electronics rely on current control loops [14,69,79]. Using these control schemes, the system can always maintain a desired level of current, regardless of changes in the load or the supply. For current control, high-precision and reliability sensors are essential components of the sensing and feedback process [80]. As a system-level example, typical photovoltaic systems include a DC/DC converter stage followed by an inverter stage (see Figure 20). Grid-tied systems require current sensors to ensure grid connection. Current sensors must measure ACs and DCs accurately but also be dynamically efficient [81]. Among the cheapest options for current sensing, non-invasive CT and HE can be used in applications where bulkiness and magnetic insertion/saturation are not important, and shunt and HE ICs can be used in applications where invasiveness is not a significant factor. However, almost none of them are suitable for HF WBG converters’ control applications.

The latest advances in embedded current-sensing technologies, such as RC, MR, and FG, can minimize or eliminate the conventional problems associated with current sensors utilized for control. According to [74], a PCB-integrated RC (as a switch-current sensor) can be easily modified to control average current and peak current, respectively, and in [77], a bidirectional current controller was implemented using a high-quality MR-based current sensor.

### 4.3. Prognostics

A condition-monitoring system is also capable of extending the lifespan of a converter by monitoring temperature, current, and other parameters, which can result in significant cost savings for power systems, as well as improved safety and reliability [14,32,82]. Furthermore, condition-monitoring systems may provide useful data for predictive maintenance and analytics. By leveraging real-time monitoring, it becomes possible for engineers to proactively detect potential issues, which can help to avoid system downtime and increase the life expectancy of a converter [31]. One of the measurement types whose information can be translated to the degradation or aging of switches or capacitors is current [83]. A high BW current sensor is typically required since most degradation-related current information is obtained from transients or HF components [84,85,86]. The aging-detection mechanism can be implemented using current information, such as switching transients, as shown in Figure 21. The inherent integrability problems associated with coaxial shunt make embedded RC and MR potentially useful for such applications. 

### 4.4. Characterization

It is common practice to use circuits for characterizing semiconductor devices, such as double-pulse testers (DPT), to characterize both static and dynamic performances by considering diverse device structures and operating conditions, which is essential for the design and optimization of semiconductor-based converters. In device characterization, switch currents are one of the measurement topics that can be used for a variety of assessment tasks, such as *R_ds-on_* measurement or the analysis of switching transients [31]. As can be seen in Figure 22, either the top or bottom switch of a leg can be considered DUT. However, since many people use (coaxial) shunt as the switch-current sensor, the low-side switch was selected as DUT to minimize isolation issues. However, invasive sensors are not the best option for WBG layouts devices, especially GaN. A non-invasive unidirectional RC switch-current sensor was used in [30] to test a SiC module for shoot-through protection, which seems promising for use in broader DPT applications. 

## 5. Discussion

Upon reviewing the information presented in the previous sections, it becomes apparent that there is no perfect single-path current-sensing method with all desirable properties that can be integrated into WBG power converters as a general solution. It is nevertheless possible to combine different single-scheme methods to achieve the desired performance in a given application, particularly when the complementary BW compensation method is used, as illustrated in Figure 23. This can range from simple current-sensing systems to advanced multi-sensory schemes. Ultimately, the selection of a sensing arrangement should be based on the specific requirements of the application. Historically, hybrid configurations have been underutilized in a variety of power electronics applications, in which two single-scheme current-measurement methods are combined to compensate for their limited bandwidth [14,38,48,49,88,89,90,91,92]. Using hybrid (or dual-scheme) configurations can increase the accuracy, speed, and BW of a sensing system, which is especially beneficial in high-frequency applications as a single measurement method may not provide enough BW [14]. 

Even though hybrid configurations may provide more viable solutions than single-path sensing systems, one of the disadvantages of hybrid configurations is that they can be more difficult to design and implement than single-path sensing systems, and they can also require complex layout considerations. Due to the combination of multiple sensing methods that can introduce errors associated with every single sub-measurement system in multi-scheme configurations, they are less reliable than simple systems. A number of integration issues can be considered when considering multi-scheme current sensors for power electronics, as previously discussed in this review. These challenges may limit the size, circuit invasion, EMI immunity, and other options of multi-scheme current sensors. Therefore, it is essential to understand how to properly select multi-scheme current sensors in order to improve the integration of these sensors into WBG power converters and maximize their performance.

## 6. Conclusions

Throughout this comprehensive review, the performance of existing single-scheme current sensors has been examined to gain a better understanding of the challenges associated with integrating these sensors into existing or emerging architectures of WBG power converters. Moreover, the potential for developing single-scheme current sensors (Figure 24) with desired performance for specific applications has been discussed. For instance, the survey evaluated the performance of single-scheme current sensors in terms of their accuracy, linearity, and dynamic range, as well as the influence of high EMI noise on their performance to be employed for protection, control, or monitoring purposes.

As shown in Table 1, current-sensing schemes are color-coded (dark green = excellent; light green = good; yellow = fair; orange = poor; and red = worst) based on their integrability concern, while in Table 2, they are scored in terms of their functionality as determined by the four main applications in power electronics systems. The types of current sensors in the last two tables are based on their market availability, implementation method, similarity of application, and their references. For specific applications with higher BW, these two tables can also be used to select single current sensors for use in combination as a hybrid current sensor.

## Figures and Tables

**Figure 1 sensors-23-06481-f001:**
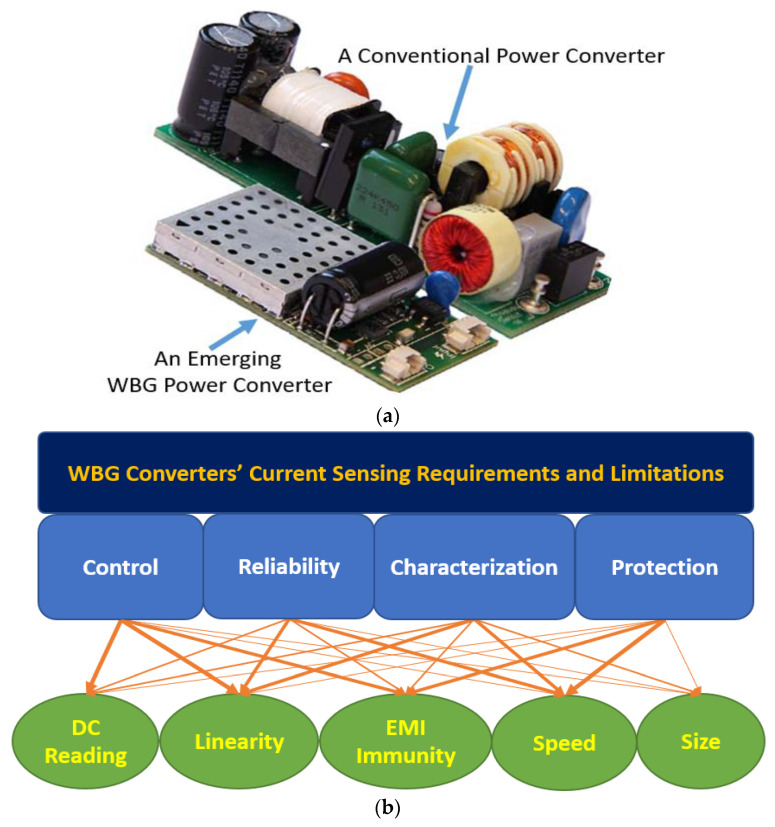
Comparison of a conventional Si-based power converter and a WBG-based converter with the same power rating [5] (**a**) and the current sensor requirements for WBG power converters (**b**).

**Figure 2 sensors-23-06481-f002:**
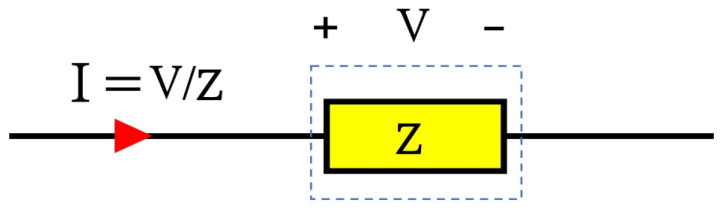
Ohmic current sensing principle.

**Figure 3 sensors-23-06481-f003:**
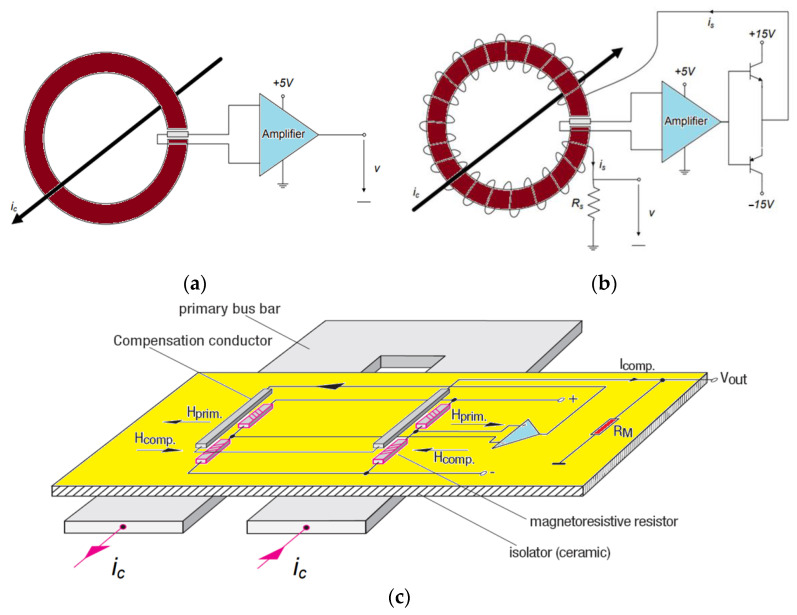
Different Hall-effect sensors’ topology: open loop with a core (**a**), closed loop with a core (**b**), and closed loop core-less (or compensated MR) (**c**) [9].

**Figure 4 sensors-23-06481-f004:**
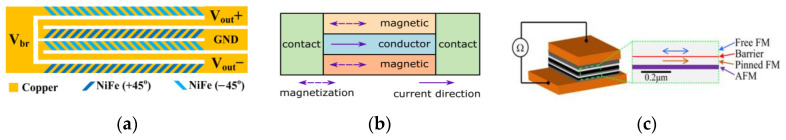
Physical structures of MR sensors: AMR (**a**), GMR (**b**), TMR (**c**) [28].

**Figure 5 sensors-23-06481-f005:**
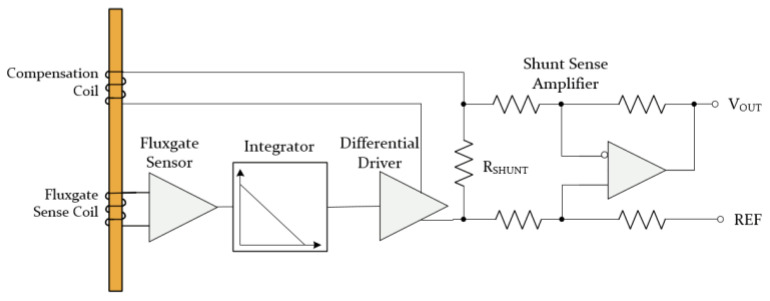
The block diagram and circuitry of a parallel fluxgate scheme.

**Figure 6 sensors-23-06481-f006:**
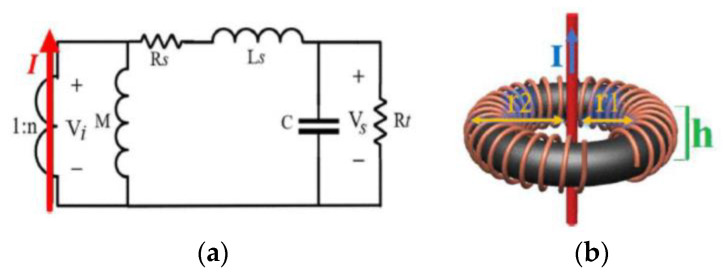
Equivalent circuit (**a**) and physical realization (**b**) of a toroidal current transformer [11].

**Figure 7 sensors-23-06481-f007:**
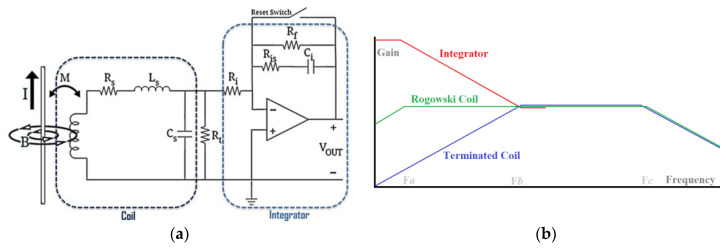
Equivalent circuit (**a**) and gain-frequency response (**b**) of a Rogowski Coil [30].

**Figure 8 sensors-23-06481-f008:**
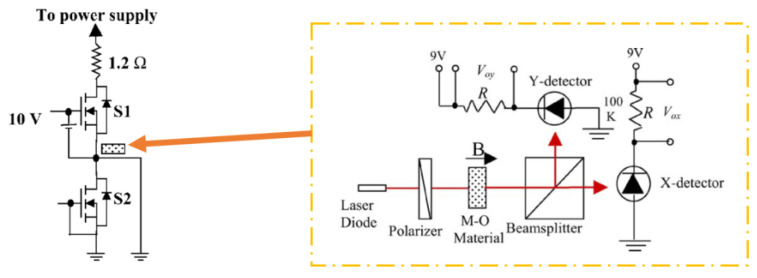
Schematic diagram of the optical sensing part placing in a leg test setup [33].

**Figure 9 sensors-23-06481-f009:**
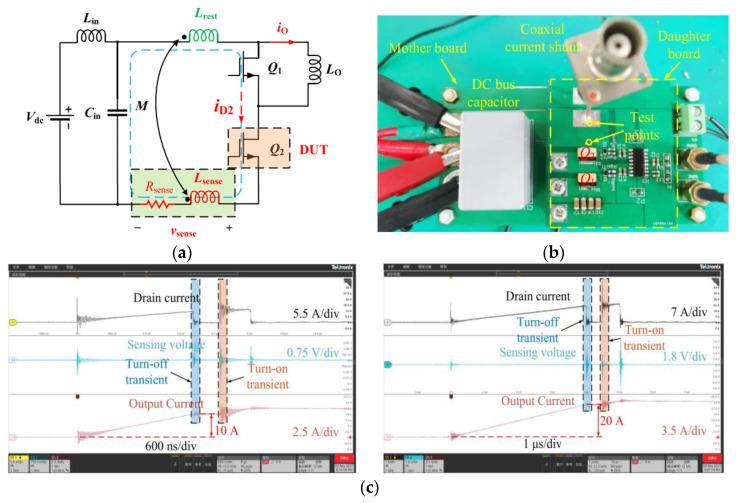
Deploying a coaxial shunt current sensor in series with low-side GaN (**a**,**b**) and its switching current reading results (**c**) [19].

**Figure 10 sensors-23-06481-f010:**
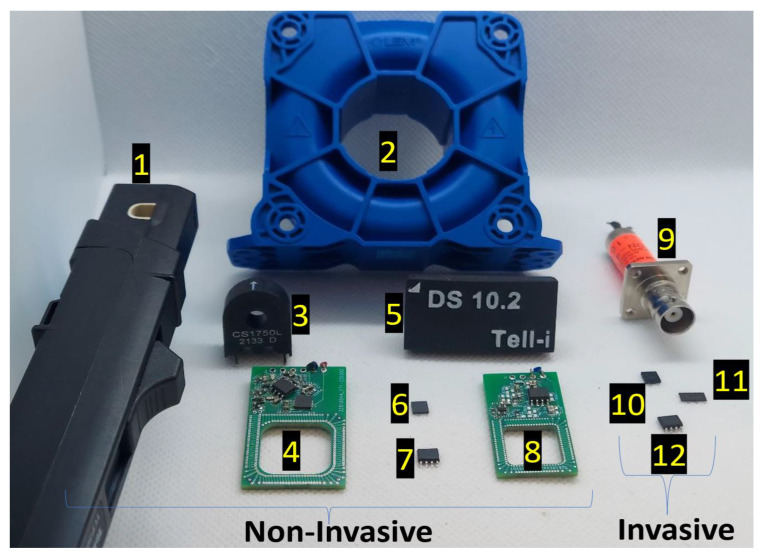
Size of different existing current sensors or probes: 1, Tektronix TCP305A; 2, LEM LF 510-S (closed-loop Hall effect); 3, Coil Craft CS1750L (current transformer); 4, UNCC’s Hybrid (Fluxgate + Rogowski); 5, Tell-i DS10.2 (Coil + MR); 6, TI DRV421 (Fluxgate); 7, HMC 1021S (AMR); 8, UNCC’s AC transducer (Rogowski Coil); 9, Coaxial Shunt; 10, Infineon TLI4970 (Hall effect); 11, SMD Shunt Resistor; 12, Allegro ACS730 (Hall effect).

**Figure 11 sensors-23-06481-f011:**
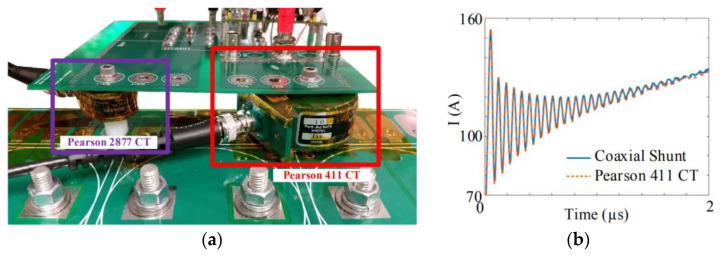
Integration of Pearson’s CTs (**a**) to monitor transient of the switching current (**b**) [40].

**Figure 12 sensors-23-06481-f012:**
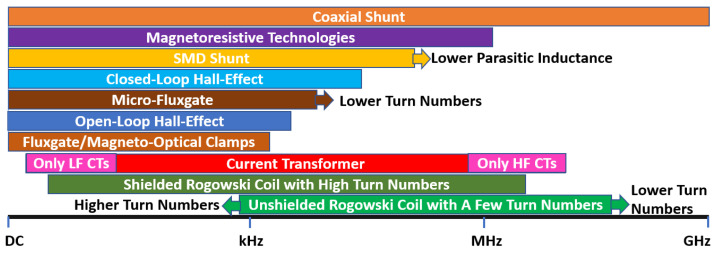
Potential bandwidth of different current-sensing methods [9,11,12,13,14,18,19,20,21,52,53,54,55].

**Figure 13 sensors-23-06481-f013:**
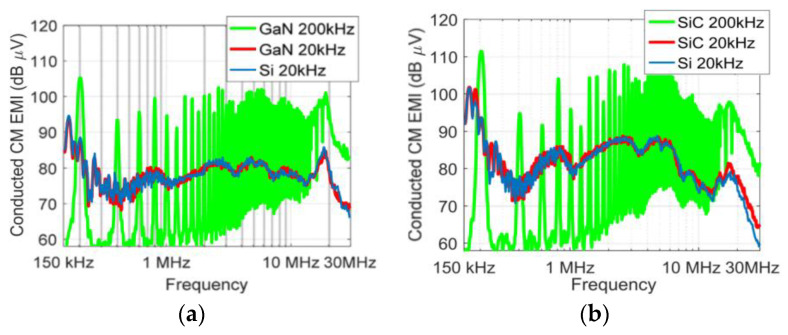
Conducted EMI in WBG drives ((**a**): GaN, (**b**): SiC) in comparison with Si-based one [56].

**Figure 14 sensors-23-06481-f014:**
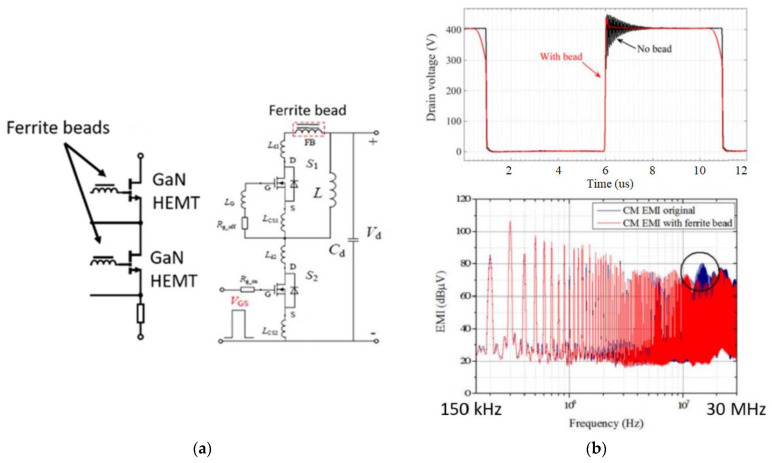
An example of layout consideration (**a**) to reduce radiated EMI (**b**) [59].

**Figure 15 sensors-23-06481-f015:**
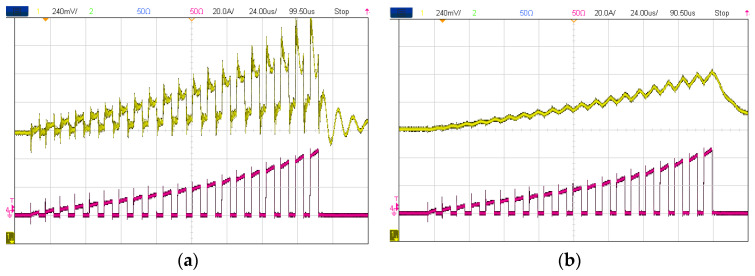
TMR’s output on a switching node: raw output (**a**) and low-pass-filtered by a passive *RC* with a 10 kHz cutoff frequency (**b**), in which the magenta waveform is the current reference measured by a DC-50 MHz probe (TCP305A [43]) and the yellow is the output of TMR2111S [65].

**Figure 16 sensors-23-06481-f016:**
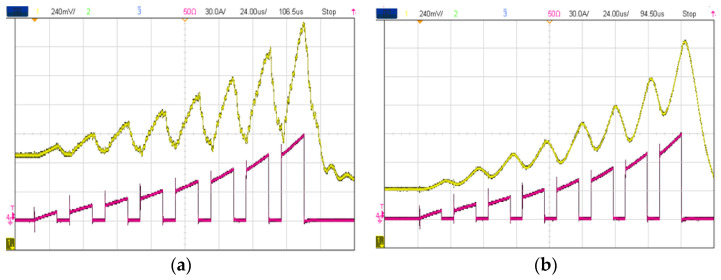
Micro-fluxgate’s output on a switching node: raw output (**a**), applied an RC low-pass at its output cutting off at 16 kHz (**b**), in which the magenta waveform is the current reference measured by a DC-50 MHz probe (TCP305A) [43] and the yellow is the output of DRV425 [46].

**Figure 17 sensors-23-06481-f017:**
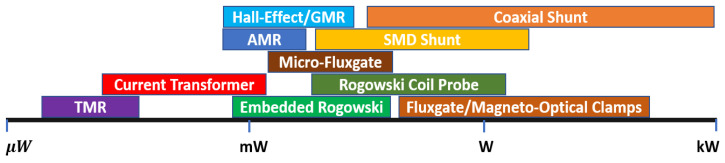
Potential power consumption of different current sensors [9,12,18,28,30,54,69].

**Figure 18 sensors-23-06481-f018:**
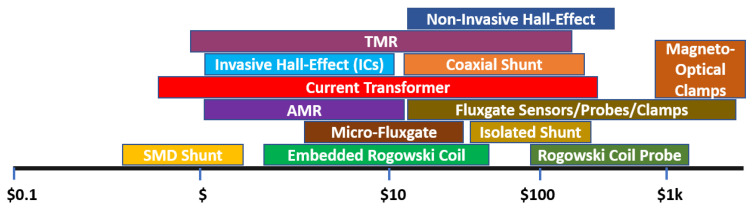
Potential cost of different current-sensing methods by Summer 2023 [71].

**Figure 19 sensors-23-06481-f019:**
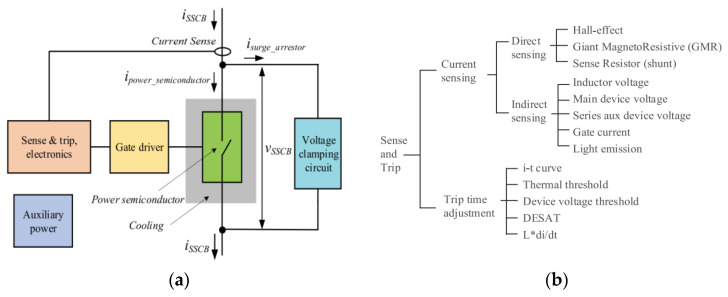
Solid-state circuit breaker: Key components (**a**); sense and trip adjustment (**b**) [75].

**Figure 20 sensors-23-06481-f020:**
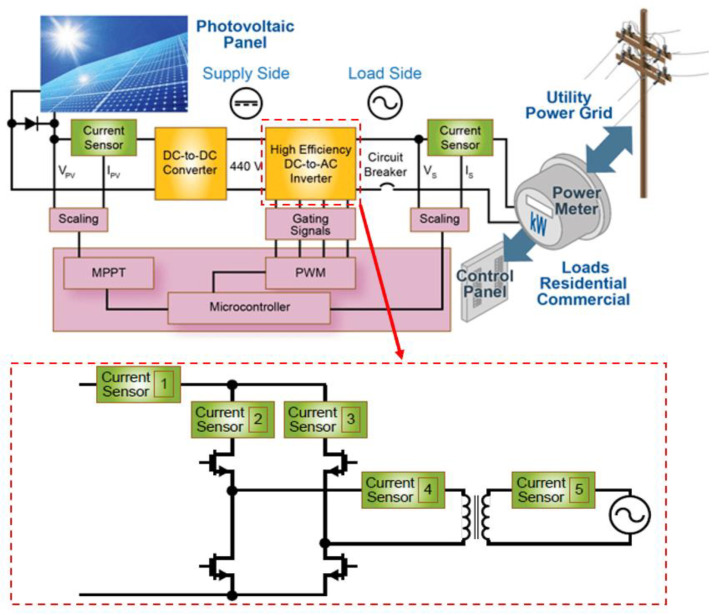
Current-sensor placement in a renewable-grid-tied system for control [81].

**Figure 21 sensors-23-06481-f021:**
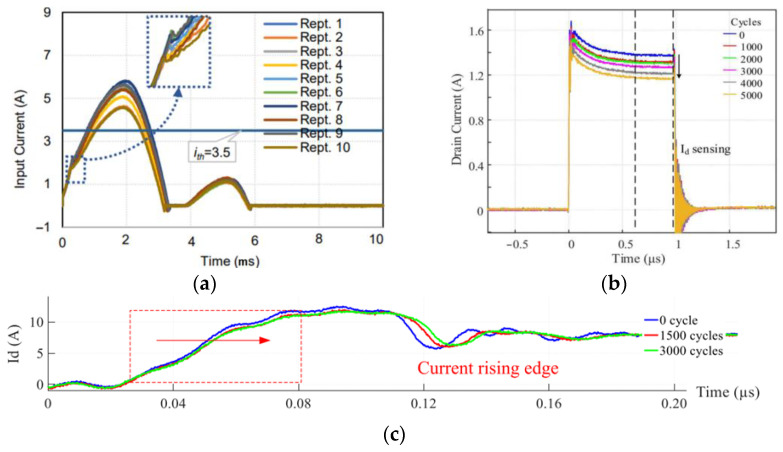
Current-based reliability assessment: DC-link capacitor [83] (**a**), Saturated SiC MOSFET [84] (**b**), and SiC in its normal operation [85] (**c**).

**Figure 22 sensors-23-06481-f022:**
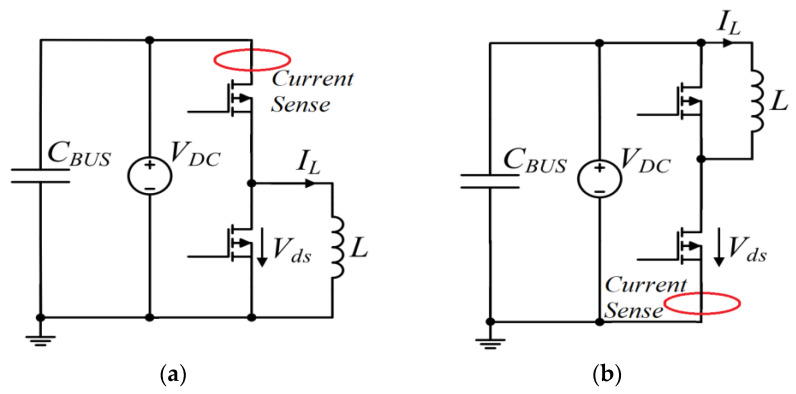
DPT testers: High-side DTU (**a**); low-side DUT (**b**) [32,87].

**Figure 23 sensors-23-06481-f023:**
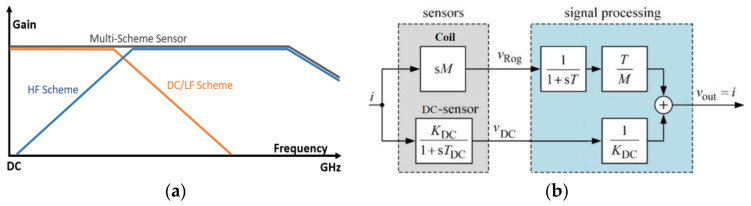
Bandwidth enhancement in dual-scheme current sensors (**a**) and Hoka principle of current sensing (**b**) [88,89,90,91].

**Figure 24 sensors-23-06481-f024:**
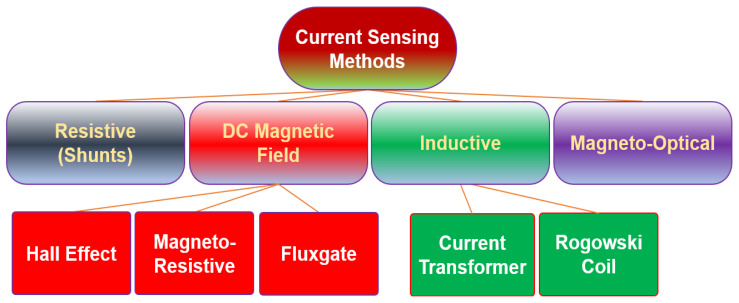
Different current-sensing methods in one picture.

**Table 1 sensors-23-06481-t001:** Current-sensing integration scores.

Scheme \Integration Concern	SMD Shunt [9,18,69]	Coaxial Shunt [19,20,21]	Isolated Shunt [35,71]	Toroidal HE [9,42,55,69]	Coreless HE [37,45,68,69]	MO Clamp [23,33,69,71]	TMR/AMR [17,28,65,71]	Toroidal FG [9,71]	Micro-FG [13,46,53,66]	RC Probe [9,71]	CT [11,69,71]	Embedded RC [12,30,32]
Isolation												
Circuit Invasion												
Size/Bulkiness												
DC Capability												
Bandwidth												
EMI Immunity												
Thermal Stability												
Accuracy												
Linearity Range												
Power Consumption												
Cost												

**Table 2 sensors-23-06481-t002:** Current-sensing application scores in WBG converters.

Application vs. Scheme	SMD Shunt	Coaxial Shunt	Isolated Shunt	Toroidal HE	Coreless HE	MO Clamp	TMR/AMR	Toroidal FG	Micro-FG	RC Probe	CT	Embedded RC
Protection												
Control												
Prognostics												
Characterization												

## Data Availability

No new data were created.

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
