# Peer review of "Current Sensor Integration Issues with Wide-Bandgap Power Converters"

_sensors, 2023, doi:10.3390/s23146481_

Round 1

Reviewer 1 Report

Please, improve the quality/legibility of graphical info throughout, especially where you use insets.

Author Response

I appreciate your thorough review. In accordance with your request, I have revised all vague information, figures, and tables. I have also added more detail and explanation to the report. I believe the changes will make it easier to understand and more useful for the audience. Please let me know if there is anything else I can do to improve it. Also, Figure 24 has been added to the conclusion for a better understanding of the classification.

Reviewer 2 Report

This paper provided a comprehensive review on all kinds of commonly used current sensors, which discussed the needs and the challenges by taking WBG power converters as an example of application. In general, the review has a practical guiding significance. Some questions need to be noted or explained more specifically.

1)    A brief introduction on WBG power Converters should be given, on its basic composition, the role and the relation of each part, how it works, typical size, typical working voltage, typical working current, etc., not just the general advantages and the applications.

2)    For Fig.1, the distinction and connection of the four parts need a more specific description. It seems like that all of the four parts have high requirements on all the five parameters. A specific range of the indications that WBG power converters needed is better to be given. The units and the size of the fonts should be unified.

3)    In part 2.1, line 114, why a magnetic core is needed in ohmic current sensors?

4)    For Fig.4, the order of (a) and (b) should be switched for a better understanding.

5)    For Fig.7 (b), what are the typical values of Fa, Fb, and Fc?

6)    There are some overlap between part 3 and part 4. Part 4 could be more brief.

7)    The types of current sensors should be defined more clearly in Table 1 and Table 2, to make the comparison more clear.

English Language is fine.

Author Response

I appreciate your thorough review. I have revised my manuscript as follows:
1) An explanation of WBG power converters has been added to the previous introduction by splitting the first paragraph into two paragraphs. Additionally, Figure 1(a) has been added for better visualization.

2) In Figure 1(b), the four parts are distinguished and linked by arrows of varying thickness, where the thicker arrow shows the greater impact of each factor based on the application.

3) As an ohmic current sensing method, one method of reading the inductor current is to read the inductor voltage and integrate it. This type of current sensing can be affected by temperature variations in the inductor core. In order to avoid confusion, the last paragraph of section 2.1 clarifies this point.

4) For Fig.4, the order of (a) and (b) has been switched for better understanding.

5) In Figure 7(b), the typical value for Fa is usually Hz~kHz, for Fb is usually 100 kHz~MHz, and for Fc is usually MHz~GHz. This information is added to the last paragraph of that subsection.

6) In parts 3 and 4, technical implementations of current sensors were considered in order to provide a better understanding. Despite the difficulty of separating parts 3 and 4, some paragraphs in part 4 have been shortened to minimize overlap.

7) The types of current sensors in the last two tables are based on their market availability, implementation method, similarity of application, and references. To clarify this, the last paragraph of the conclusion has been extended. Also, Figure 24 has been added to the conclusion for a better understanding of the classification.

Reviewer 3 Report

This paper investigates existing single-scheme current sensors' performance to understand the challenges associated with incorporating them into the architecture of emerging WBG power converters. In addition, the potential of developing novel methods to improve the performance of these single-scheme current sensors is also explored. The research is detailed and the expression is smooth. My questions/suggestions are as follows:

1.       Selecting multi-scheme current sensors in order to improve the integration of these sensors into WBG power converters may not be a good solution, such as increasing the cost and volume of the system, the difficulty of integration.

2.       There are more than three colors in Tables 1 and 2, and the ranking of scores should be explained in detail.

3.       Many images in this paper are not numbered with (a)(b)(c), as shown in Figures 9 and 11, 15, and 16, which contain several images.

Author Response

I appreciate your thorough review. I have revised my manuscript as follows:
1) Although hybrid (multi-scheme) current sensors may require greater implementation complexity as well as be larger and more expensive, they could be used in applications in which bandwidth and accuracy are more important than size and cost. For instance, hybrid current probes (such as TCP305A from Tektronix) or trace current sensors (such as Tell-i 10.2) are appropriate examples.

2) The colors in Tables 1 and 2 are ranked as follows: Dark Green represents excellent, Light Green represents good, Yellow represents fair, Orange represents poor, and Red represents worst. In addition, each column contains sufficient references to clarify the accuracy of the ranking. Also, Figure 24 has been added to the conclusion for a better understanding of the classification.

3) The captions and numbering of all images have been corrected accordingly.